# Enhancing Cement Paste Properties with Biochar: Mechanical and Rheological Insights

Daniel Suarez-Riera [1], Luca Lavagna [2,*], Juan Felipe Carvajal [1], Jean-Marc Tulliani [2], Devid Falliano [1] and Luciana Restuccia [1,*]

1. Department of Structural, Geotechnical and Building Engineering, Politecnico di Torino, Corso Duca degli Abruzzi 24, 10129 Turin, Italy; daniel.suarez@polito.it (D.S.-R.); juan.carvajal@studenti.polito.it (J.F.C.); devid.falliano@polito.it (D.F.)
2. Department of Applied Science and Technology, Politecnico di Torino, Corso Duca degli Abruzzi 24, 10129 Turin, Italy; jeanmarc.tulliani@polito.it
* Correspondence: luca.lavagna@polito.it (L.L.); luciana.restuccia@polito.it (L.R.)

**Abstract:** Biochar, the solid sub-product of biomass pyrolysis, is widely considered an effective water retention material thanks to its porous microstructure and high specific surface area. This study investigates the possibility of improving both mechanical and rheological properties of cement pastes on a micro-scale. The results show that using biochar as a reinforcement at low percentages (1% to 5% by weight of cement) results in an increase in compressive strength of 13% and the flexural strength of 30%. A high fracture energy was demonstrated by the tortuous crack path of the sample at an early age of curing. A preliminary study on the rheological properties has indicated that the yield stress value is in line with that of self-compacting concrete.

**Keywords:** biochar; cement-based composites; mechanical properties; rheological properties; fresh-state; sustainability





## 1. Introduction

The world population grew from 1 to 8.1 billion from the year 1800 onwards, while more than 50% of the population lives in urban areas [1], and this percentage will continue to grow in the future. For these reasons, today, Portland cement concrete is by far the most produced and consumed construction material in the world [2–4]. The composition of cement has evolved since the patent of Joseph Aspdin in 1824, and nowadays, concretes should satisfy different requirements like improved workability, strength and ductility, durability, and sustainability. To meet these challenges, specific admixtures, new filler/aggregates, nanomaterials, new curing conditions, etc., have been explored [5–9]. Engineering materials' properties are governed by their microstructure. Currently, micro- and nano-carbon reinforcement have been shown to improve the performances of cement-based composites—for example, carbon nanotubes, graphene and carbon fibers [10–13]—showing significant improvements in mechanical, thermal, and electrical properties. However, even if carbon-based reinforcement significantly influences the mechanical properties of the composites, their use has a high cost and environmental impact [14]. In addition, carbon-based reinforcements are difficult to disperse in the cementitious matrix homogeneously, due to their hydrophobic nature [15]. The use of biochar as a carbon-based reinforcement can improve the mechanical properties of cementitious materials, while at the same time being economically sustainable and environmentally friendly, mitigating $CO_2$ emissions that contribute to the greenhouse effect [16–19]. The use of biochar as a carbon reinforcement within cementitious matrix composites has received particular attention in recent years [20–29]. Ahmad et al. [30] observed that the addition of 0.08 wt. % coconut shell-derived biochar improves cement paste's mechanical properties, such as compressive strength and fracture toughness by 25% and 76%, respectively. The presence of inhomogeneous biochar particles

in the tensile plane of cement paste samples was shown to affect the rupture modulus, reducing it by 20–40%. In this article, the authors highlight the importance of particle size because the interaction between the biochar and cement particles plays an essential role in resisting crack propagation and absorbing fracture energy. Some of the authors in a previous article [31] demonstrated that hazelnut shell biochar used as a micro aggregate improves the cement paste's mechanical properties. The results suggest that adding a low percentage of biochar (0.8–1 wt. %) improves cement paste's modulus of rupture and fracture energy by 22.2 and 61%, respectively. The latter is attributed to the crack path's tortuosity, due to the introduction of biochar particles that induce the matrix's inhomogeneity and attract the crack path towards them. Furthermore, biochar derived from exhausted coffee powder is a promising low-cost carbon-based material for improving the mechanical behavior of cement-based composites [16]. Suarez-Riera et al. [32] discovered that integrating 1% and 2% of biochar composites enhanced the flexural strength of cement paste by up to 24% and 15%, respectively. As demonstrated by some of the authors, biochar was also used for improving the fracture energy of mortars [33].

Similarly, Choi et al. [34] reported that the addition of biochar in mortar at low percentages improves the compressive strength by about 10%. Moreover, the authors investigated the use of biochar as a cement replacement in cementitious mortars with improved results, for low percentages of biochar. Nonetheless, when using 5, 10, and 15% biochar as a replacement in cement, the capability of mortar to flow was reduced by 3, 17, and 20%, respectively. Gupta and Kua [35] also discussed the potential of using biochar as a building material to capture and "lock" atmospheric carbon in civil infrastructure. Micropores are associated with the adsorption of liquid and gas compounds; therefore, retaining water in a cement matrix helps the optimal formation of cement hydrated phases [17,35–37]. In this work, the influence of biochar's addition, a commercial pyrolyzed carbon-based material from wood waste, as a reinforcement in cement-based composites was studied. The mechanical properties of the samples, like their fracture energy, as well as their flexural and compressive strength, were then determined. Also, preliminary analyses of the mixture's behavior in the fresh state, such as regards its shear stress, yield stress, and viscosity, after adding different percentages of biochar were investigated and discussed.

## 2. Materials and Methods

### 2.1. Materials

The cement composite samples were made using a type I cement 52.5 R purchased from Buzzi-Unicem, Roccavione, Italy. According to the cement manufacturer, the cement had a Blaine specific surface area of 0.4–0.55 $m^2/g$, an initial setting time > 90 min, a specific gravity of 3.14 $g/cm^3$ and a mean particle size of about 16 $\mu$m [38]. The composition of the cement is given in Table 1. Biochar (B) (NERA Company s.r.l, Torino, Italy) was employed in the sample's preparation. Superplasticizer (SP) provided by Master Builders Solutions, Treviso, Italy, was used to reduce the water-to-cement ratio (w/c) and achieve good workability and biochar particle dispersion in the mix. Biochar was ground for 7 h using the ball milling method in a ceramic jar with agate balls. Then, the ground biochar was sieved for 30 min using an ASTM mesh 80 (180 microns) sieve employing a short-period oscillatory movement produced by the compact vibration sieve. This procedure was carried out because previous research indicated that the smaller the average size of the biochar particles, the better the performance of the cement composites, as the surface area of the particles will increase. For B, the reinforcement percentages added to the mix by weight of cement were 1, 2, 3, 5 and 7%. These percentages were chosen to verify the possibility of using higher percentages of biochar as a reinforcement after the excellent results obtained from former studies [30–33,39].

**Table 1.** Composition of the used cement as reported by the supplier.

| Oxide | $SiO_2$ | $Al_2O_3$ | $Fe_2O_3$ | CaO | MgO | $SO_3$ | $Na_2O$ | $K_2O$ | Loss on Ignition |
|---|---|---|---|---|---|---|---|---|---|
| Amount (%) | 20.02 | 4.12 | 1.87 | 63.23 | 4.2 | 3.43 | 0.003 | 0.0015 | 0.8 |
| Phase | $C_3S$ | | $C_2S$ | | $C_3A$ | | | $C_4AF$ | |
| Composition | 49.10% | | 19.70% | | 7.91% | | | 5.20% | |

After the milling process, the biochar was characterized by means of laser granulometry (Malvern Mastersizer 3000, Malvern, UK with Aeros S dry cell measurement), X-ray diffraction (XRD, Pan'Analytical X'Pert Pro, Eindhoven, the Netherlands, Cu K$\alpha$ anticathode, $\lambda = 0.154056$ nm) and X-ray fluorescence (XRF, Rigaku NEX CG II, Cedar Park, TX, USA). Field emission–scanning electron microscopy (FE-SEM, Hitachi S3800, Tokyo, Japan) was used to observe the microstructure of the powder.

*2.2. Methods*

The water retention capacity was measured following the method suggested by Gupta et al. [17]: two biochar samples, each of 10 g, were placed in a beaker, then put in a ventilated oven at 90 °C for 24 h in order to remove the moisture that was adsorbed. Then, each of the samples was filled with 100 g of distilled water and left to stand for 48 h in each beaker. A vacuum filtration test was prepared, in which a fritted glass funnel (G4) was used to remove the solid phase from the liquid one. The fluid retention capacity was calculated by taking the wet biochar weight and subtracting the dry one. The thermal stability was assessed using a thermogravimetric analyzer (TGA, Mettler Toledo 1600) in air condition. The air was supplied at a constant flow rate (50 mL min$^{-1}$). A roughly 33 mg sample was placed in an alumina crucible and the sample was heated from 25 °C to 1000 °C with a constant heating ramp of 10 °C min$^{-1}$.

Regarding mechanical tests, 40 prismatic specimens of 20 mm × 20 mm × 80 mm were prepared. For each experimental set, eight specimens were made, with four tested after 7 days of curing and the remaining four tested after 28 days (Table 2). The flexural tests were conducted according to the ASTM C 348, while for compressive strength the procedure of ASTM C 109 was followed. Regarding the rheological test activity, a total of 12 mixed specimens were made, 2 for each experimental set (Table 2).

**Table 2.** Set of experimental samples.

| | Mechanical Test Activity | | Rheological Test Activity |
|---|---|---|---|
| ID | N° Specimens 7 Days | N° Specimens 28 Days | N° Specimens |
| Ref. OPC | 4 | 4 | 5 |
| B 1% | 4 | 4 | 5 |
| B 2% | 4 | 4 | 5 |
| B 3% | 4 | 4 | 5 |
| B 5% | 4 | 4 | 5 |
| B 7% | - | - | 5 |

The specimens were prepared following the mix design shown in Table 3. Ground biochar was dispersed in a solution of water and superplasticizer. The solution was transferred into a mixing bowl and hand-mixed for 1 min until yielding a homogeneous solution. The cement was gradually poured into the solution in the first two minutes while operating the mixer at 480 rpm for three minutes. Then, the mixer speed was increased to 840 rpm for three more minutes, reaching a total mixing time of six minutes.

**Table 3.** Material compositions of pastes tested for mechanical and rheological properties.

| I.D. | Mechanical Test Activity | | | | Rheological Test Activity | | | |
|---|---|---|---|---|---|---|---|---|
| | Cement (g) | Water (g) | B (g) | SP (g) | Cement (g) | Water (g) | B (g) | SP (g) |
| Reference OPC | 460 | 161 | - | 4.6 | 50 | 17.5 | - | 0.5 |
| B 1% | 460 | 161 | 4.6 | 4.6 | 50 | 17.5 | 0.5 | 0.5 |
| B 2% | 460 | 161 | 9.2 | 4.6 | 50 | 17.5 | 1.0 | 0.5 |
| B 3% | 460 | 161 | 13.8 | 4.6 | 50 | 17.5 | 1.5 | 0.5 |
| B 5% | 460 | 161 | 23.0 | 4.6 | 50 | 17.5 | 2.5 | 0.5 |
| B 7% | - | - | - | - | 50 | 17.5 | 3.5 | 0.5 |

The samples subjected to tests for rheological activity were immediately loaded into a co-axial cylinder rotary viscometer for testing (Malvern Kinexus Pro+, Malvern, UK). Two different measurements were taken at 23 °C—steady shear and dynamic oscillatory shear, with an angular frequency ω of 6.28 rad/s (ω = 2πf with f, frequency = 1 Hz). Five minutes of temperature stabilization was allowed after the pastes were introduced into the cup, while the gap between the base of the cup and the bob was set to 5 mm. Successively, the samples used for mechanical tests were transferred to the prismatic stainless-steel molds and we left them to cure for 24 h at room temperature, covered to maintain the level of humidity at 90%. After this initial curing, the samples were demolded, and put into a water tank at 20 °C for curing up to the mechanical tests carried out at 7 and 28 days. Before the test, the specimens were notched employing a cutting machine BRILLANT 220 with a 2 mm-thick diamond cut-off wheel, making a 6 mm deep U-notch, following the Japan Concrete Institute Standard JCI-S-001 [40].

The methodology use for rheology testing is reported in the supporting information. Notched hardened specimens were subjected to a three-point bending test (TPB test) with a single-column Zwick Line-Z050 testing machine with a 1 kN load cell device and the clip-on strain gauge to measure the crack mouth opening displacement (CMOD). The span (distance between the supports) adopted was 65 mm, and a test speed of 0.005 mm/min was set. The evaluation of the flexural strength was done according to Equation (1):

$$\sigma_{f\ max} = F_{max} \cdot \frac{3L}{2bh^2} \quad [\text{MPa}] \tag{1}$$

where $L$ is the span equal to 65 mm, $b$ is the specimen depth, and $h$ is the net ligament height equal to 20 mm and 14 mm, respectively. The evaluation of fracture energy ($G_F$) was performed according to the Japan Concrete Institute Standard JCI-S-001 [40] (Equation (2)).

$$G_F = \frac{0.75W_0 + W_1}{A_{lig}} = G_{F0} + G_{Fcorr} \quad \left[\text{N/mm}^2\right] \tag{2}$$

where $A_{lig}$ is the area of the nominal ligament equal to 280 mm², $W_0$ [N·mm] is the area below the CMOD curve up to the rupture of specimen and $W_1$ [N·mm] is the work done by the deadweight of the specimen and loading, calculated according to Equation (3):

$$W_1 = 0.75\left(\frac{l}{L}m_1 + 2m_2\right)g \cdot CMOD_c \quad [\text{N·mm}] \tag{3}$$

where $l$ is the loading span (distance among the supports), equal to 65 mm, $L$ is the total length of specimen, equal to 80 mm, $m_1$ (kg) is the mass of the notched specimen, $m_2$ (kg) is the mass of the loading arrangement not attached to the testing machine but placed on the beam until rupture, $g$ is the gravity acceleration and $CMOD_c$ is the crack mouth opening displacement at the rupture. The compression test was performed employing the same testing machine, changing the load cell capacity to 50 kN and setting a displacement rate

equal to 0.5 mm/min, utilizing the two broken prisms from the TPB test. The compressive strength is computed as follows:

$$\sigma_{c\ max} = \frac{F_{max}}{bh} \quad [\text{MPa}] \tag{4}$$

where $F_{max}$ is the maximum force supported by the specimen before rupture, and $b$ and $h$ are the specimen thicknesses (20 mm on either side).

### 2.3. Biochar Characterization

#### 2.3.1. Laser Granulometry

Figure 1 shows the biochar particle size distribution after grinding and sieving. About 90% of the biochar particles are below 24 µm, the average particle size is equal to 7.9 µm, and about 10% of the biochar particles are below 1.8 µm. In comparison with the used cement powder's particle size distribution, a significant portion of the biochar's particles are finer than the cement's ones. The low dimension of biochar can allow the biochar to be dispersed between the cement powder particle, thus biochar can act as a nucleation point of the hydrated phase of cement, as demonstrated by Ling et al. [41].

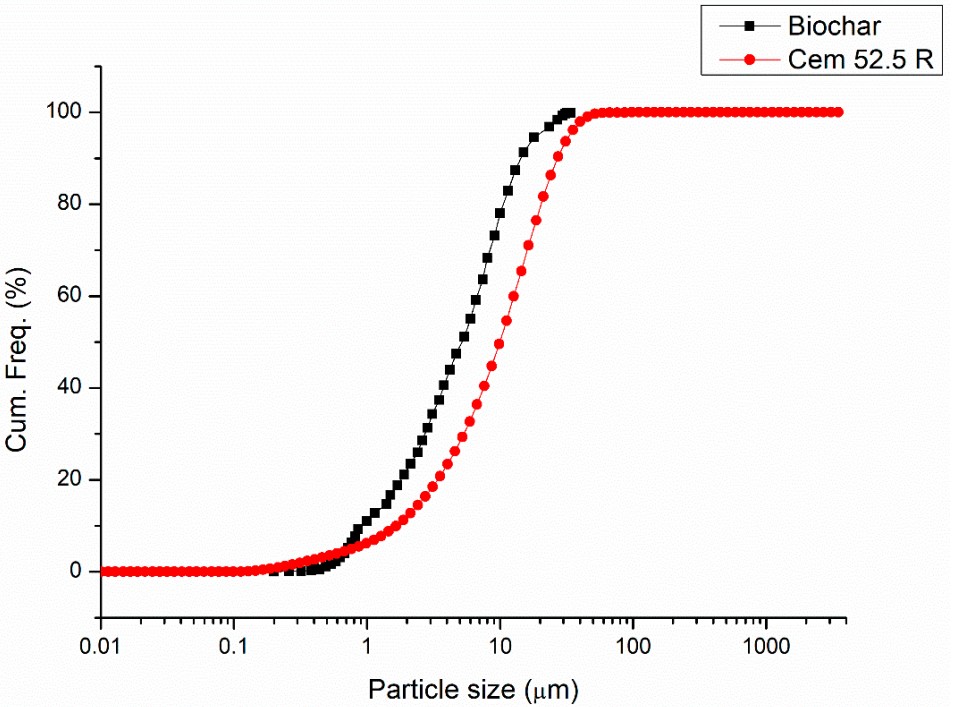

**Figure 1.** Particle sizes distribution of biochar after the milling process and cement used (52.5 R).

#### 2.3.2. X-ray Diffraction

Figure 2 shows that the ground material is mostly amorphous (seen from the broad hump centered at about 24° in 2θ, probably relative to the amorphous carbon), with some peaks of calcium carbonate (JCPDS card number 05-0586) and quartz (JCPDS card number 41-1045). However, both quartz and calcium carbonate are present in traces.

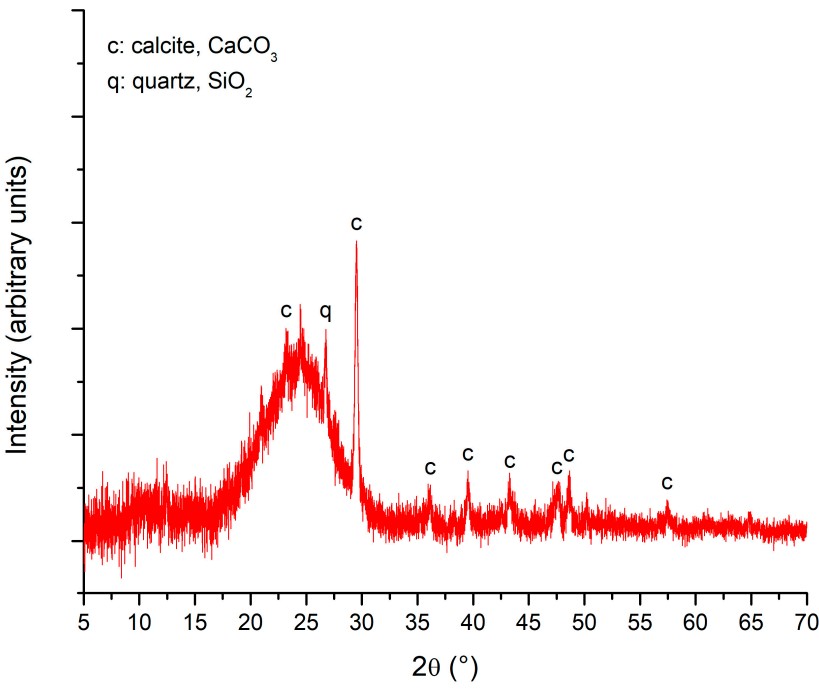

**Figure 2.** XRD pattern of ground biochar.

### 2.3.3. X-ray Fluorescence

The biochar was mainly made of carbon (97.4 wt. %), with traces of CaO (1.43 wt. %), $K_2O$ (0.43 wt. %), MgO (0.26 wt. %), $P_2O_5$ (0.12 wt. %) and $SiO_2$ (0.10 wt. %). These results are in line with the XRD results reported in Figure 2.

### 2.3.4. Field Emission Scanning Electron Microscopy (FESEM)

The biochar particles' morphology is shown in Figure 3a,b. Figure 3a shows the structure of wood-chip biochar particles, where some surrounding particles have a honeycomb pore structure. These particles do not appear to have excess pores, which thus prevents water absorption and excellent saturation, making them less effective than other biochars analyzed in the literature concerning the retention of liquids and gases [42]. On the other hand, in Figure 3b, a pore surface can be observed in the size range of 5–10 microns, where a series of smaller particles are evidenced within these cavities in a range between 2 and 4 microns or smaller. In the same image, the structure of the smallest pores can be observed, and it can be noted that this structure is probably that of a softwood; this is strictly linked to the feedstock used as biomass. Figure 3c,d show that grinding mostly destroyed the macro-porous structure of biochar particles, decreasing their liquid and gas retention capacities. The size of the observed particles is coherent with laser granulometry measurements. Lehmann [43] suggests that pores smaller than 30 μm are more effective in retaining water; however, Shafie et al. [44] agree with this theory, but explain that the best efficiency is obtained with a pore diameter of 5–6 μm, which enables better fluid retention. Finally, pore diameters ranging from 10 to 30 μm tend to absorb more water in the mix, lowering the free water content for cement hydration and workability [45].

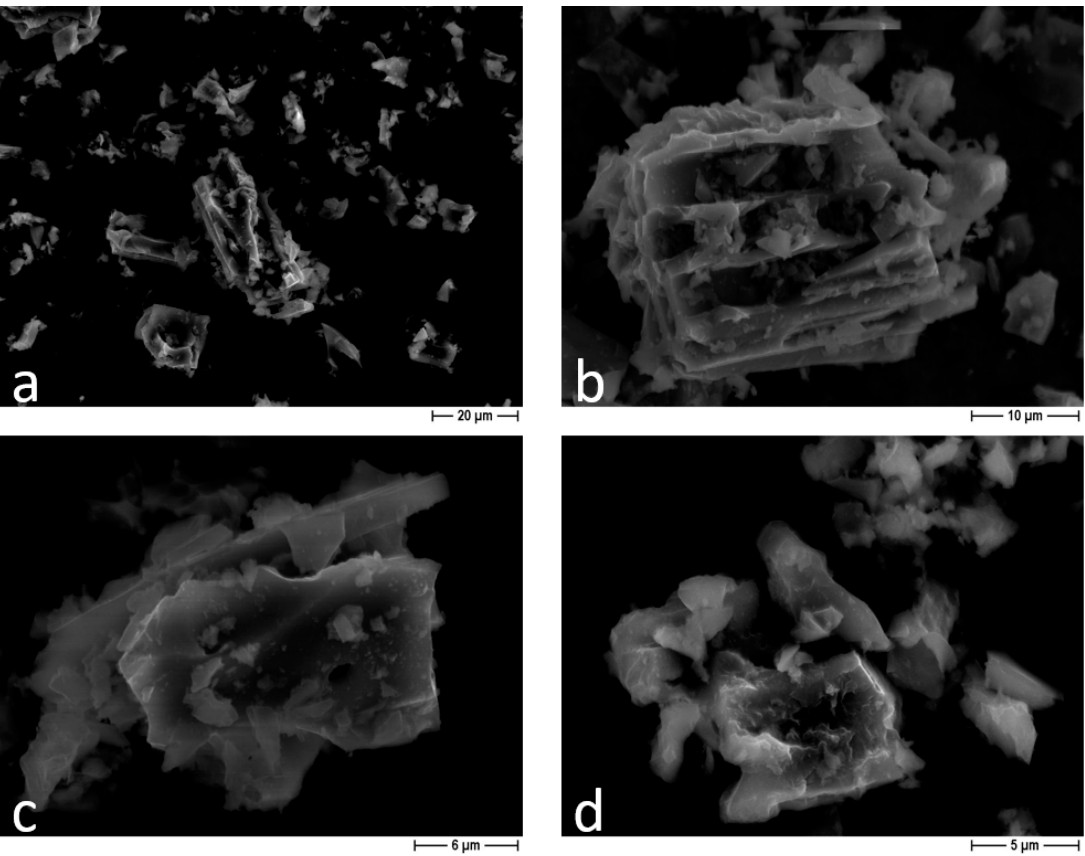

**Figure 3.** FE-SEM micrograph: 7 h ground and sieved biochar. (**a**) ×1000; (**b**) ×2500; (**c**) ×4000; (**d**) ×5000.

### 2.3.5. Water Retention Capacity

The water retention capacity was determined (as the mass of absorbed water per gram of dry biochar) as 0.94 ± 0.02 g/g. The water retention capacity here was predicted to be higher than those of other biochars previously examined, such as the biochar produced by the pyrolysis of mixed wood sawdust utilized by Gupta et al. [17], where the liquid retention capacity was 2.5 ± 0.2 g/g. On the other hand, Suarez-Riera et al. [32] used the same method to evaluate the water retention capacity proposed in this article, obtaining a 2.17 g/g water retention for the Gray Borgotaro biochar. In fact, it is found that ground biochar has a much lower water retention capacity than other types of biochar due to the low porosity after the grinding process, which damaged the former plant tracheids. The water uptake capacity of ground biochar is compatible with FESEM observations, which showed that the original porous structure was only partially maintained after the grinding step.

### 2.3.6. Thermogravimetric Analysis (TGA)

The TGA curve shown in Figure 4 reveals two main thermal phenomena: first, a weight loss (more than 5%) from 50 to 130 °C due to the desorption of adsorbed water in the biochar pores, which is in accordance with Zhou et al. [46], followed by a second weight loss that starts above 270 °C and continues until 480 °C with a wight loss of 90%, which is attributable to the decomposition of the carbonaceous residue, leaving a final residue at 1000 °C of 5% of ash. The TGA results align with the XRF ones, where the C content was estimated to be about 97%.

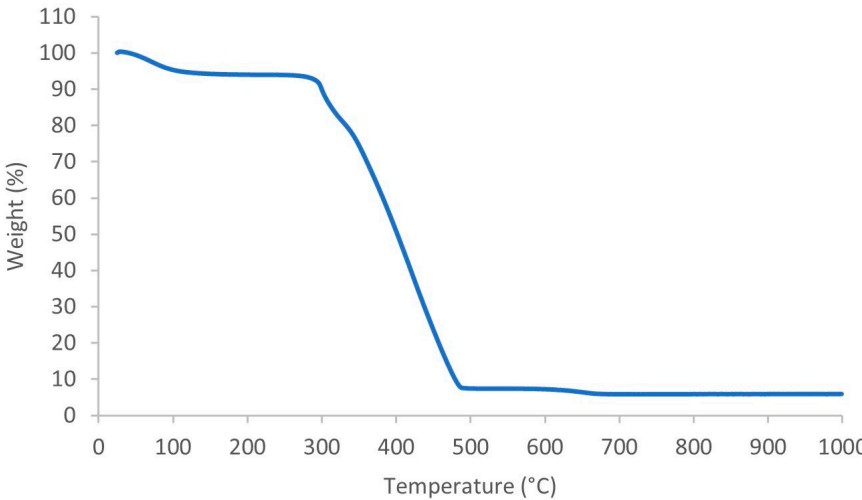

**Figure 4.** Thermo-gravimetric analysis of biochar under air atmosphere.

### 3. Results

#### 3.1. Effect of Biochar Addition on Flow Curves

The flow curves reported in Figure 5 show that the higher the biochar addition, the higher the shear stress for any fixed value of the shear rate. They were then fitted using a least squares function corresponding to the Bingham model. A good linear fit ($R^2$ values > 0.99) between the shear stress and shear rate was obtained for a shear rate range from 4 to 200 s$^{-1}$ (Table S1).

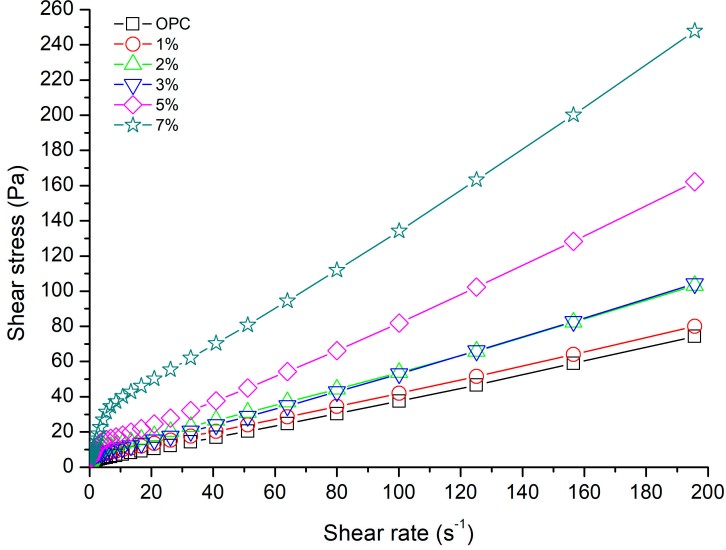

**Figure 5.** Flow curves of the investigated compositions.

From Figure 6, we see that, regardless of the degree of biochar addition to the cement paste, when the shear rate increases, the viscosity decreases, and then remains relatively constant, with a negligible shear thickening behavior. The shear thickening behavior is due to the high hydrodynamic (lubrication) forces between monodispersed spherical particles overcoming the particle repulsion forces that formed temporary aggregates [47]. The results regarding the Bingham model and yield stress are reported in the supporting materials.

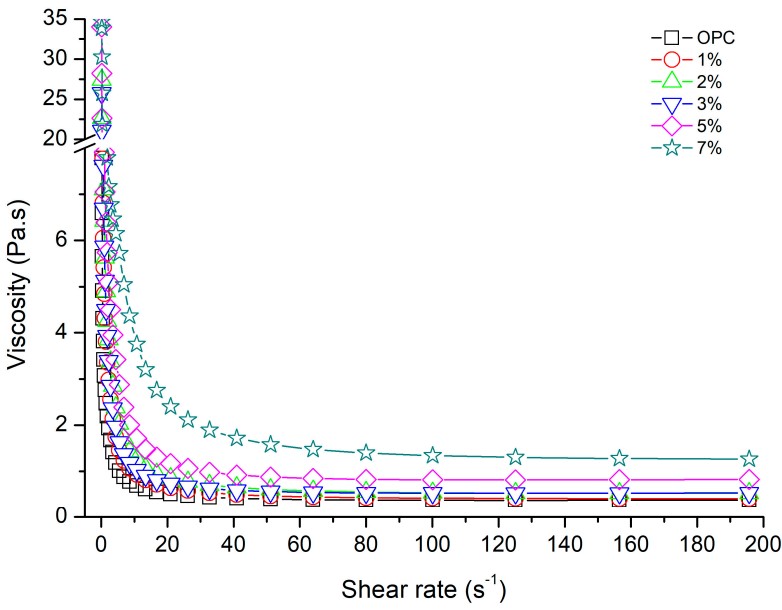

**Figure 6.** Viscosity as a function of the shear rate.

Figure 7 shows the results of dynamic oscillatory shear tests using strain sweep. Due to the higher viscosity of the sample with 7% biochar, it was not possible to perform the measurement on this sample. The storage modulus, G′, represents the deformation energy stored by the sample, while the loss modulus, G″, represents the lost dissipated deformation energy. At low amplitudes (linear viscoelastic range, LVE), the elastic behavior dominates over the viscous one in all samples (G′ > G″). This is typical of a (weak) gel structure with a certain stability. When the limit of the LVE is reached, the structure of the gel is altered (irreversible deformations) [48]. The limits of the LVE (calculated for a 5% deviation from the linear region of G′ curves) were about 0.024% for the pure cement paste and 0.019%, 0.019%, 0.018% and 0.016%, respectively for 1, 2, 3 and 5% biochar additions to the cement pastes. Table 4 reports the initial (at 0.01% strain) storage and loss modulus values.

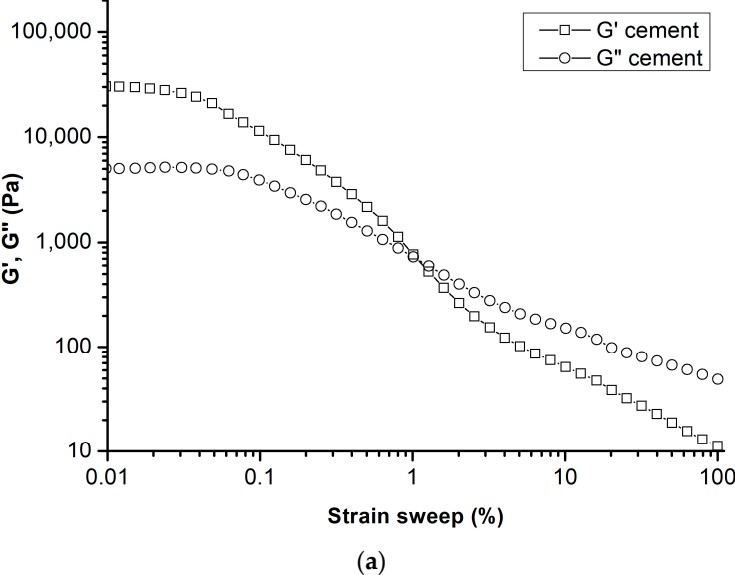

(**a**)

**Figure 7.** *Cont.*

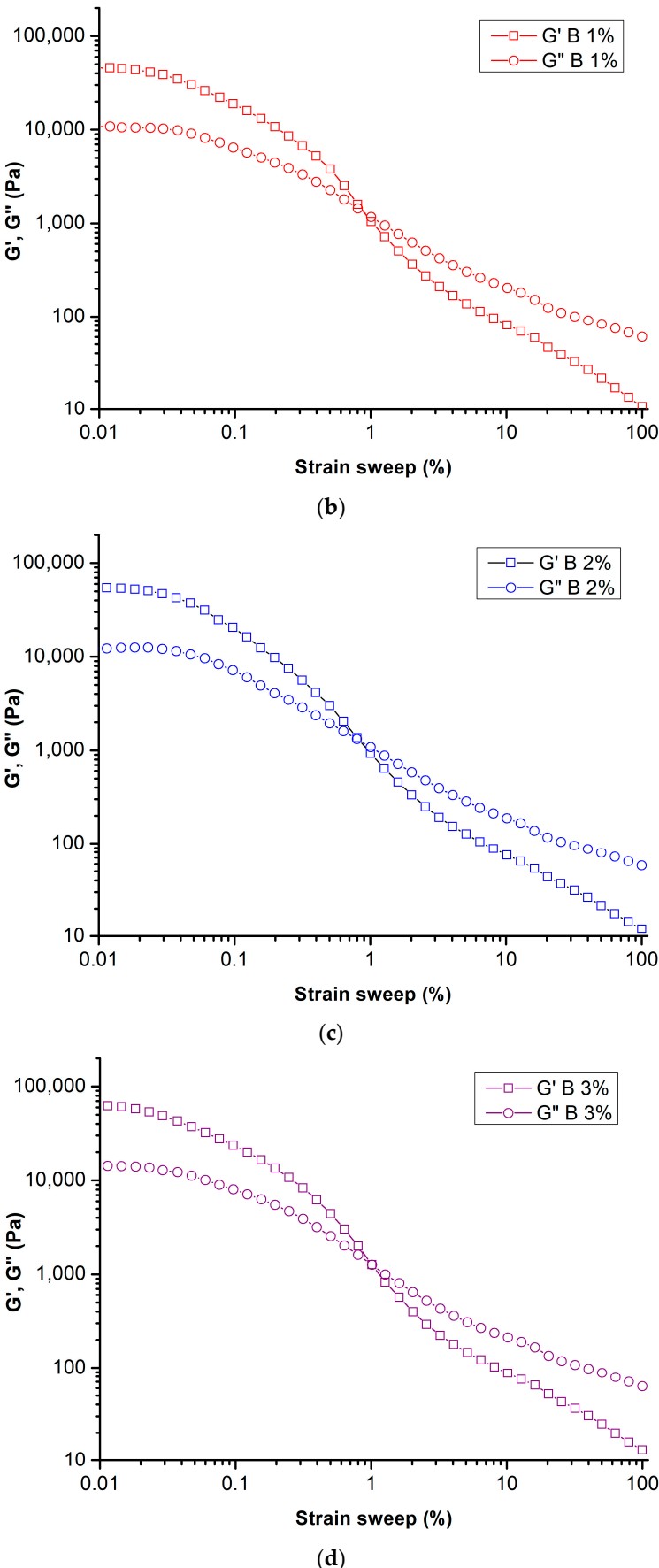

**Figure 7.** *Cont.*

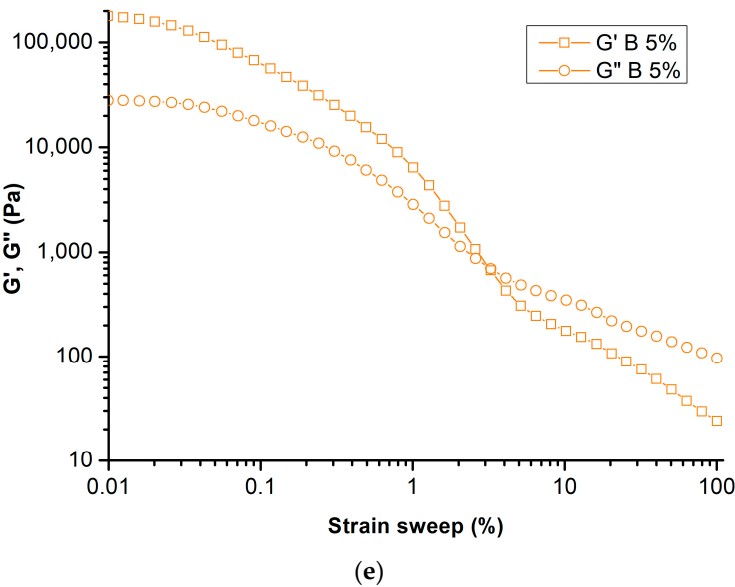

(**e**)

**Figure 7.** Dynamic oscillatory shear measurement using strain sweep: (**a**) cement paste, (**b**) cement paste + 1% B, (**c**) cement paste + 2% B, (**d**) cement paste + 3% B, (**e**) cement paste + 5% B.

**Table 4.** Initial storage and loss modulus value.

|  | Storage Modulus (Pa) | Loss Modulus (Pa) |
|---|---|---|
| Reference OPC | 30,340 | 5034 |
| B 1% | 45,790 | 10,760 |
| B 2% | 54,050 | 12,250 |
| B 3% | 63,380 | 14,440 |
| B 5% | 181,300 | 29,360 |

### 3.2. Mechanical Properties

The mechanical test results related to the 7 days of curing are shown in Figure 8A, showing that adding microparticles to the cement mix significantly improves the mechanical properties of plain cement. Regarding flexural strength, the most effective additions were B 1% and 5%, while compressive strengths were 1% and 2%, respectively. Besides the fracture energy, the most effective percentage of B was 3%. However, adding 1% of B achieved an almost 130% improvement with respect to the reference mixture. Results regarding the mechanical properties related to 28 days of curing are shown in Figure 8B, and they are not as promising as those of 7 days. In this case, the best performance for flexural and compressive strength was seen in B with the addition of 2%, yielding improvements of 30% and 13% with respect to the reference sample, respectively. A similar trend was observed in the work of Suarez-Riera et al. [32]. Besides this, evaluating the fracture energy, we found that a biochar content of 5% presented the best performance. Furthermore, the B contents for additions of 1 to 3 wt. % of cement showed the same positive performance with respect to those samples without biochar.

Figures 9–12 show how the fracture surface changes when biochar is added to the cement matrix. The use of biochar alters the crack path within the cementitious matrix, thereby creating a winding crack path, consequently increasing the fracture energy.

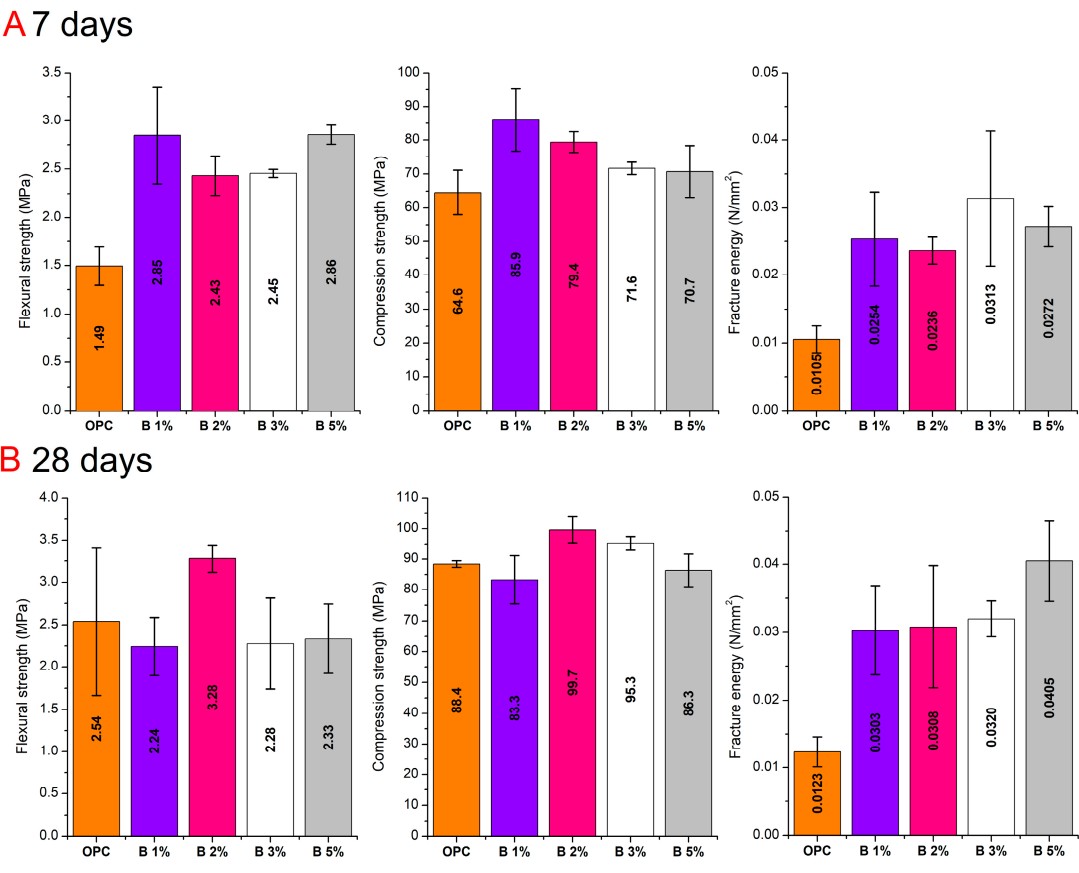

**Figure 8.** Mechanical properties of cement composite containing biochar (**A**) after 7 days of curing and (**B**) after 28 days of curing.

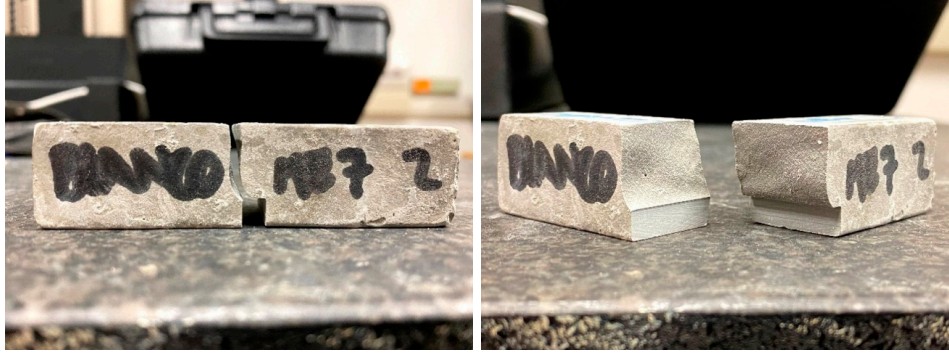

**Figure 9.** Linear crack path and smooth fracture surface. Plain cement (OPC) at 7 days.

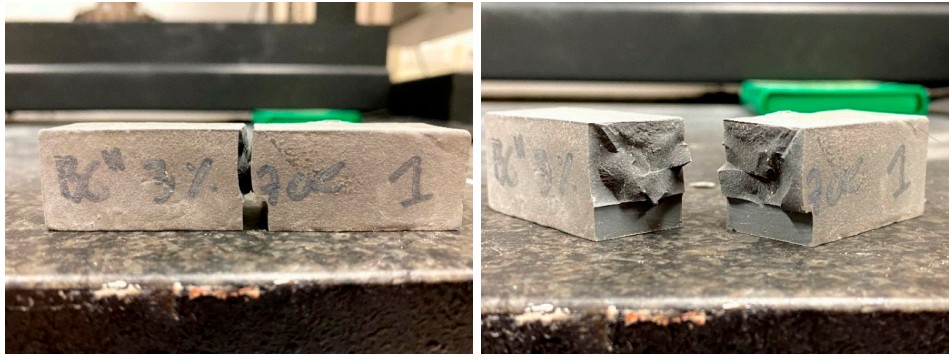

**Figure 10.** Pseudo-linear crack path and rough fracture surface. B 3% at 7 days.

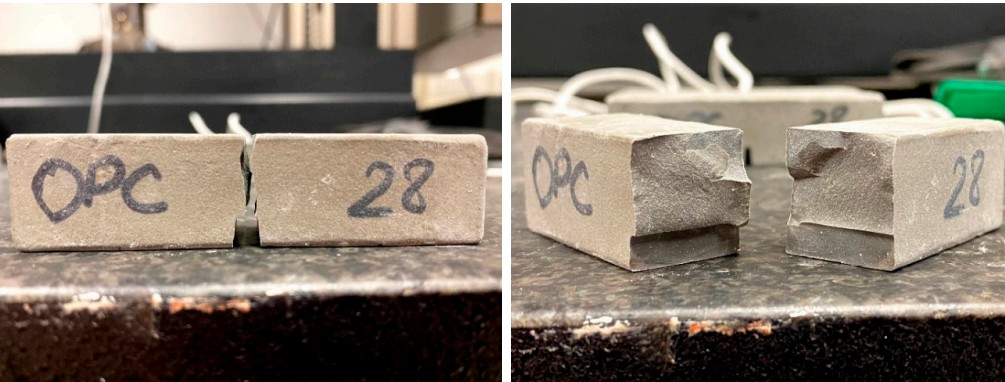

**Figure 11.** Linear crack path and pseudo-smooth fracture surface. Plain cement (OPC) at 28 days.

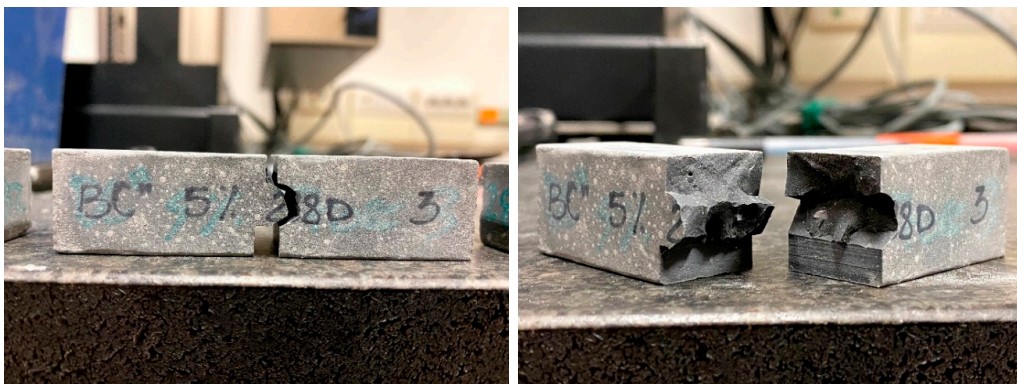

**Figure 12.** Tortuous crack path and rough fracture surface. B 5% at 28 days.

## 4. Discussion

As shown in the Supplementary Materials, when stress is applied to the system of cement–biochar, if this stress is higher than the cohesion forces of the agglomerates, the water trapped within the agglomerates is released, thus lowering the viscosity. In this study, biochar can increase both yield strength and viscosity. The results indicate that the yield stress value is in line with that of self-compacting concrete. Dynamic oscillatory shear tests using strain sweep showed that the higher the biochar addition, the lower the LVE. Considering that biochar particles are smaller than cement ones (Figure 1), this behavior is also probably due to the limited chemical affinity between biochar and the cement gel. In addition, the sharp edges of biochar particles (Figure 3) contribute to gel structure alteration. Thus, when a reduced strain is applied to the system, the gel structure is rapidly destroyed. The initial storage modulus value increased with biochar content (Table 4), indicating that the elasticity was higher, as well as the stiffness, in agreement with steady shear measurements.

The energy dissipation due to mechanical vibrations is produced by calcium silicate hydrates (C–S–H) sliding at different scales. At the nanoscale, the C–S–H sheets slide with the interlayer water acting as the lubricating layer to dissipate energy. At the microscale, gel pores are present and C–S–H globules can also slide with capillary pore water as the lubricating layer, leading to further energy dissipation [49]. Biochar addition increased the initial loss modulus (Table 4), probably because of the higher interfacial friction with C–S–H sheets.

From the mechanical point of view, biochar is a good reinforcement for cement-based composites. As already investigated [16,32,50,51], the addition of small amounts of B can increase the flexural strength of cementitious materials. The strength of 7-day cementitious composites depends on the densification of the matrix: the higher the packing density, the higher the mechanical properties. Furthermore, biochar can act as a starting point for hydration products' growth [52]. As a result, B samples have higher mechanical properties

than pristine cement samples at early ages. However, the increase in mechanical strength is not as pronounced as in the case of other carbon-based reinforcements used in the literature. This is also because, as known, the dispersion of a nano–micro reinforcement within the matrix also affects the mechanical outcome. In the literature, dispersion has a fundamental role in enhancing the interaction between the reinforcement and the matrix [16,53–55]. The literature suggests that a proper dispersion of the nano- and microparticles in the final mix is essential to achieving a homogeneous material, as this is directly proportional to the particle size: the smaller the particle size, nearer to the nanoscale, the greater the surface area per unit volume [56]. This leads to increased van der Waals forces and electrostatic force among nanoparticles, which greatly favor their re-agglomeration [55]. As seen in Figure 3, biochar particles have a wood chip structure because of the type of biomass used for the production. This structure allows a good cohesion and a good interaction with the cementitious matrix, thus guaranteeing a bond with the matrix, which explains the flexural and compressive strength enhancement [42,44]. On the other hand, the biochar absorbs water, and it is possible that the increase in mechanical properties at 28 days, especially with large amounts of water, is due to the reduction in the w/c ratio, which consequently leads to an improvement in mechanical properties. From the visual comparison of fracture surfaces, it is observed that in the case of samples cured for 7 days, the presence of biochar (Figure 10) created a much more tortuous path compared to standard cement (Figure 9). As the curing time increased up to 28 days, it can be noted that the fracture surface of the cement maintained the same surface characteristics (Figure 11), while with the presence of biochar (Figure 12), it showed an increased tortuous path, thus causing an increase in the fracture energy required to break the specimens.

## 5. Conclusions

This work explored the effects of incorporating biochar particles, resulting from industrial pyrolyzed wood biomass processes, as a filler in standard cement paste composites. The biochar particles showed excellent compatibility with the cement matrix without encountering interphase separation, thanks to the porosity of the biochar evidenced by the FE-SEM observations. From the results obtained in terms of flexural and compressive strength and fracture energy, it can be concluded that the addition of biochar as a reinforcement improves the mechanical properties of cement paste. By analyzing the overall nature of the tested specimens, the biochar percentage of 2% seemed to be the optimal one for improving mechanical properties while containing the increase in the viscosity of the mix. This interesting result, associated with the highest fracture energy and comparable mechanical strengths with respect to the reference samples, represents the starting point for a forthcoming study related to the use of biochar as a green filler for cement-based composites. In particular, future studies will explore the possibility of using biochar within mortars as a specific substitute for natural aggregates or cement.

**Supplementary Materials:** The following supporting information can be downloaded at: https://www.mdpi.com/article/10.3390/app14062616/s1, methods for rheology test; Table S1: Summary of the main rheological parameters; Figure S1: Yield stress in function of plastic viscosity and biochar content. References: [51,52,57–62].

**Author Contributions:** Conceptualization, D.F. and L.R.; methodology, D.S.-R., J.-M.T. and L.L.; software, D.S.-R. and J.-M.T.; validation, L.L., D.F. and L.R.; formal analysis, J.F.C. and D.S.-R.; investigation, D.S.-R., J.F.C. and J.-M.T.; resources, L.R. and L.L.; data curation, D.S.-R. and D.F.; writing—original draft preparation, D.S.-R., J.F.C. and D.F.; writing—review and editing, L.R., L.L., J.-M.T. and D.F.; visualization, D.S.-R. and L.L.; supervision, L.R., D.F. and L.L.; project administration, L.R.; funding acquisition, L.L. All authors have read and agreed to the published version of the manuscript.

**Funding:** This work was supported by Master Builders Solutions and Nera Biochar SRL.

**Institutional Review Board Statement:** Not applicable.

**Informed Consent Statement:** Not applicable.

**Data Availability Statement:** Dataset available on request from the authors.

**Acknowledgments:** The authors gratefully acknowledge the interdepartmental laboratory SISCON (Safety of Infrastructures and Constructions) of Politecnico di Torino for use of instruments (laser granulometry and rheometer).

**Conflicts of Interest:** The authors declare no conflicts of interest.

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
