# Peer review of "Enhancing Cement Paste Properties with Biochar: Mechanical and Rheological Insights"

_applsci, doi:10.3390/app14062616_

Round 1

Reviewer 1 Report

Comments and Suggestions for Authors

The review of the paper entitled “Enhancing Cement Paste Properties with Biochar: Mechanical and Rheological Insights” by Daniel Suarez-Riera, Luca Lavagna, Juan Felipe Carvajal, Jean-Marc Tulliani, Devid Falliano and Luciana Restuccia submitted to the “Applied Sciences” journal (applsci-2913608).

 The paper concerns the possibility of improving mechanical and rheological properties of cement pastes by using porous biochar particles as "micro reinforcement". The manuscript presents original research and results. The manuscript appear to be suitable for publication in its current form after some minor corrections and explanations. Some remarks (including editorial notes) are listed below.

Remarks and editorial notes:

1) In the titles of subsections 2.1.2 -2.1.4, it would be worth expanding the abbreviations of individual methods to their full names.

2) The first paragraph of section 2. “Materials and Methods” contains data from the technical data sheets of individual materials. Please add the source of the producer/manufacturer data sheet data in the reference list.

3) Lines 88-89 say: “1, 2, 3, 5% and 7%, deriving from former studies [16,17,20,29–32]. Please briefly explain how these specific percentage levels were selected on the basis of the earlier studies mentioned - why such values?

4) As for information given between lines 109-110 and later between lines 117-122 – were the specimens prepared according to procedures described in any specific standards? If so, please provide the number and title of such a standard, otherwise write that it was done according to the authors' method and provide details of specimen preparation and test procedure.

The same in the case of all testing procedures in this section – please provide the numbers and titles of standards describing the applied testing procedures or write that the particular test was done according to the authors' method.

5) Lines 107, 108, 127, 243, 244, 245 – unnecessary spaces between the temperature value and the Celsius degree symbol – please check and correct the manuscript in this regard.

6) Lines 111, 112 - unnecessary spaces between the bracket and “Table…” – please check and correct the manuscript in this regard.

7) Line 194 – typing/numbering mistake (the text says “Figure 23”).

8) Please remove the Figure 1 as it does not add much information other than to state that the tested prism had a cut in the middle. The scheme of the 3-point bending test is understandable and it is also described in the text.

9) Line 222 – there should be “±” instead of “+”

10) Line 240: the unnecessary enter and some of the text jumped to the next line.

11) Delate the sentence written between lines 248-250 as it does not add any information to the manuscript.

12) In Table 3 caption instead of “Amount of material required for each type of test” wouldn’t be better to write: “Material compositions of pastes tested for mechanical and rheological properties”? Also please explain in the table caption the BC and SP symbols.

13) In scientific paper there should be no commercial names of the used materials, unless it was necessary for some particular reason. Please remove the commercial names – e.g. in case of cement, biochar and superplasticizer.

14) Please remove Table 4 and Table 5. They repeat the information from Figure 8 and do not add anything to the content of the manuscript. However, in Figure 8 please add the values of individual properties and the spread or standard deviation.

15) Lines 292, 316 – unnecessary spaces in the text.

16) Figures 9-12 are not mentioned in the text – please explain the differences on the particular Figures. Also Figure 12 should be Figure 11.

Comments on the Quality of English Language

The English language is good, although the text could be slightly improved in some cases, e.g. the Table 3 caption.

Reviewer 2 Report

Comments and Suggestions for Authors

The paper " Enhancing Cement Paste Properties with Biochar: Mechanical 2 and Rheological Insights " describes the importance of using biochar as reinforcement to improve the performance of cementitious materials in fresh and hardened state. The topic studied by the Authors has a significant importance and clearly written and relevant for readers of Applied Sciences journal. Therefore, my recommendation is to accept the publication with major revisions, which I believe would enhance the manuscript for the readers.

Major comments:

1. Research by the authors that was previously conducted on this scientific problem is mentioned in the article text. But a more detailed description of previously conducted studies and specification of how the current study differs from previously performed ones is lacking in “Introduction”.

2. What is the typical particle retention rate in liquid of the cellulose filters used in this study?

3. The Materials and Methods section should be improved to make it more understandable and easier to follow. Subsections and figures should be added.

4. I would like to see a more detailed description and discussion of the results obtained in the fresh state of cement paste. Additional rheology tests must be performed such as strain sweep and time sweep. The results collected from these two tests are very important to understand the effect of biochar on the behavior of cement in the fresh state.

5. The water retention capacity of biochar used in this study is 0.94 ± 0.02 g/g. Alternatively, despite the addition of 5% biochar, there was no variation in the quantity of water added for cement hydration. Consequently, the available water for cement hydration decreased, resulting in a lower W/C ratio. This suggests that the observed outcomes in this study could be attributed to a lower W/C ratio. Hence, it may be prudent for the authors to thoroughly deliberate on this aspect in their discussion.

6. It's essential to explore the potential results when replacing biochar with finely ground sand or limestone filler in this study.

7. Further directions for the scientific research development should be reflected in the “Conclusions” section.

Minor comments:

1. Line 194: Figure 2

2. Line 292: Figure 8 to Figure show

3. Line 308: I would like to see a more detailed description of this results

Round 2

Reviewer 2 Report

Comments and Suggestions for Authors

Comments and recommendations were taken into account in the article revised version. The article may be accepted for publication in its current edition.